# Dancing for Parkinson’s Disease Online: Clinical Trial Process Evaluation

**DOI:** 10.3390/healthcare11040604

**Published:** 2023-02-17

**Authors:** Meg E. Morris, Victor McConvey, Joanne E. Wittwer, Susan C. Slade, Irene Blackberry, Madeleine E. Hackney, Simon Haines, Lydia Brown, Emma Collin

**Affiliations:** 1Academic and Research Collaborative in Health (ARCH), La Trobe University, Melbourne, VIC 3086, Australia; 2Victorian Rehabilitation Centre, Healthscope, Glen Waverley, VIC 3150, Australia; 3Fight Parkinson’s, Surrey Hills, Melbourne, VIC 3086, Australia; 4Physiotherapy Department, La Trobe University, Melbourne, VIC 3086, Australia; 5Physiotherapy Department, Monash University, Melbourne, VIC 3086, Australia; 6CERI and John Richards Centre for Rural Ageing Research, La Trobe University, Melbourne, VIC 3086, Australia; 7Department of Medicine, Division of Geriatrics and Gerontology, Emory University School of Medicine, Atlanta, GA 30329, USA; 8Atlanta Veterans Affairs Center for Visual and Neurocognitive Rehabilitation, Decatur, GA 30033, USA

**Keywords:** Parkinson’s disease, rehabilitation, physiotherapy, dance, exercise, physical activity, digital healthcare

## Abstract

**Background:** Dancing is an engaging physical activity for people living with Parkinson’s disease (PD). We conducted a process evaluation for a PD trial on online dancing. **Methods:** “ParkinDANCE Online” was co-produced by people with PD, healthcare professionals, dance instructors, and a PD organisation. The evaluation mapped the following inputs: (i) stakeholder steering group to oversee program design, processes, and outcomes; (ii) co-design of online classes, based on a research evidence synthesis, expert advice, and stakeholder recommendations; (iii) trial fidelity. The key activities were (i) the co-design of classes and instruction manuals, (ii) the education of dance teachers, (iii) fidelity checking, (iv) online surveys, (v) and post-trial focus groups and interviews with participants. The outputs pertained to: (i) recruitment, (ii) retention, (iii) adverse events, (iv) fidelity, (v) protocol variations, and (vi) participant feedback. **Results:** Twelve people with PD, four dance instructors and two physiotherapists, participated in a 6-week online dance program. There was no attrition, nor were there any adverse events. Program fidelity was strong with few protocol variations. Classes were delivered as planned, with 100% attendance. Dancers valued skills mastery. Dance teachers found digital delivery to be engaging and practical. The safety of online testing was facilitated by careful screening and a home safety checklist. **Conclusions:** It is feasible to deliver online dancing to people with early PD.

## 1. Introduction

Implementing research-informed therapeutic interventions into clinical practice is a cornerstone of evidence-based clinical practice [1,2,3]. For chronic conditions, such as Parkinson’s disease (PD), it is important to implement high value, individually tailored therapies co-designed with end-users [4,5,6,7,8]. Wide variations exist in motor and non-motor symptoms of PD [9,10], and the rates of progression vary markedly. Because Parkinson’s is multi-dimensional, yet currently has no cure, a comprehensive care model is advocated [6,11,12]. Comprehensive care incorporates best-practice medical, nursing, and allied health interventions, as well as evidence-based complementary therapies [13] and exercise [14,15]. Structured physical activities play an important role in enabling people with PD to maintain fitness, health, and wellbeing [16,17,18].

Dance therapy is one form of structured physical activity thought to be beneficial for some people living with Parkinson’s disease [19,20]. It is gaining support world-wide and has growing evidence of therapeutic benefit [21,22,23,24,25]. A recent systematic review [26] found dance to be an accessible and enjoyable form of exercise for people with PD, with particular benefit to mobility and wellbeing in the early stages of disease progression. Dancing affords opportunities for creative expression and social engagement, in addition to increasing exercise participation and assisting people to move well, despite movement disorders and non-motor symptoms [27,28,29].

Dance classes for people living with PD have been implemented in community settings, either as group tuition or one–one sessions [28,30,31,32]. Recently there has also been a global shift towards the online delivery of dancing for PD [33]. The COVID-19 pandemic [34] and systematic reviews [35,36,37,38,39] have drawn attention to the benefits of access to digital modes of structured physical activity for people living with chronic neurological conditions [40,41,42,43].

“ParkinDANCE” is an evidence-based dance therapy program for people with Parkinson’s disease. It has been co-produced and implemented by Fight Parkinson’s (Australia), implementation scientists, and stakeholders from the Parkinson’s community, including people living with Parkinson’s disease [29]. Key elements of ParkinDANCE include rhythmical, music-cued movements, and dance routines designed by dance instructors, in collaboration with physiotherapists. Each ParkinDANCE class has a warm-up phase, active dance component for 30–45 min, and a cool down phase. The active dance phase affords therapeutic benefits from visual and auditory cueing [44], attention strategies [45], strengthening [14,46], and aerobic exercise [47]. The mix of dance genres and dance routines used in each class, together with music selections, vary according to individual needs and preferences.

Driven by the restrictions associated with the COVID-19 pandemic and the requirement for many people to stay at home, we co-created “ParkinDANCE Online” and conducted a clinical trial [29]. Because it was a new and complex intervention, we mapped each stage of design and implementation against our project logic model, using a process evaluation. The US Medical Research Council process evaluation framework [48,49] was used to evaluate the inputs, activities, and outputs for the digital delivery of dancing for people living with Parkinson’s disease. The specific aims were to assess (i) the contextual factors that shaped how ParkinDANCE Online was delivered, (ii) the interventions and procedures delivered, (iii) whether ParkinDANCE Online was implemented as planned, as well as the extent to which the expected outputs actually occurred, and (iv) the stakeholder views on implementation success and outcomes and their early feedback on whether the overall goals were achieved, what barriers were encountered, and what changes might be needed.

## 2. Materials and Methods

We registered on the ANZ Clinical Trials Registry (ACTRN12620001042932) and the La Trobe University Human Ethics Committee granted ethics approval (HEC18520). We have also provided open access Appendix A containing the process mapping model and resources supporting trial implementations that were evaluated in the process evaluation. This includes assessor guidelines (Appendix A), program logic model (Appendix A), participant guide (Appendix A), manual for dance instructors (Appendix A), and video conference guide for the trial (Appendix A).

### 2.1. Design

A process evaluation addressed each of the aims, based on the model by Moore et al. [49] and the Medical Research Council framework for process evaluations [48]. In the context of a clinical trial, a process evaluation analyses whether predetermined research program activities have been implemented as planned and whether they resulted in particular outputs. According to French et al. (2022) [48], process evaluations are very helpful to include in clinical trials because they enable a greater understanding of the mechanisms of complex interventions by investigating implementation processes, the context of trial delivery, and the mechanisms of impact. They yield data that can help interpret outcomes, as well as assisting the implementation of findings into future clinical practice (French et al., 2022) [48].

See Appendix A for the overall process logic model, with Figure 1 highlighting details for implementation of the ParkinDANCE Online project.

### 2.2. Participant Eligibility

Stakeholders in this evaluation were people with idiopathic PD (dancers), dance instructors, physiotherapists (who were assessors and process auditors), occupational therapists, nurses, and representatives from Fight Parkinson’s, Victoria, Australia. A research analyst developed and administered a Research Electronic Data Capture (REDCap) database [50] and accompanying surveys, with input from the project steering committee.

People living with PD were included if they met the following inclusion criteria: modified Hoehn and Yahr stages 0–2.5 [51]; aged 18–65 years; judged by a medical practitioner to be able to perform exercises at home safely and without hands on assistance; and able to walk 10 metres. Those with cognitive impairment or neurological, musculoskeletal, or cardiopulmonary conditions that precluded safe independent practice were excluded. Participants needed to have internet access and a digital device, such as a laptop or smartphone.

The dance instructors were experienced in conducting dancing classes for people with basal ganglia conditions. They had been trained by MEM and VM in the pathophysiology of movement disorders and management of PD. As with the health professionals in the project, they had completed good clinical practice (GCP) training, first aid training, and education in safe and effective delivery of digital dance sessions.

The physiotherapists, occupational therapist, and nurse were registered with the Australian Health Practitioner Regulating Authority. Fight Parkinson’s was the key industry partner and is a not-for-profit organisation that provides education, advice, advocacy, and health programs for people with PD and their families.

### 2.3. Recruitment of PD Dancers

The PD dancers were recruited by Fight Parkinson’s using convenience sampling [52]. Fight Parkinson’s followed a co-designed recruitment protocol and placed advertisements for the trial in their “In Motion” magazine, on their website, and on pamphlets distributed at support group meetings. People who were interested in participating telephoned the Fight Parkinson’s hotline and were screened for suitability by a movement disorders nurse. If they met all screening criteria and signed informed consent and a medial release form, their medical practitioner was asked to complete a brief assessment to confirm that they were safe and eligible to participate.

### 2.4. Participant Characteristics

Twelve people with comparatively early-stage PD, four dance instructors and two movement disorders physiotherapists, were included in the 4-week trial. There were nine females and three males with a mean age of 58.1 (SD: 8.1) years and an age range of 34 to 63 years included in the dancer group. The mean duration of PD was 4.3 (SD: 3.7) years, with the mean Hoehn-Yahr PD rating scores of 1.8 (SD: 0.7). Seven participants were retired, five worked part- or full-time and three participants reported falls within the last 6-months. Their demographic details are summarised in Table 1.

There were four dance instructors (three females and one male) with a mean age of 43.6 years and 18.6 years of dance teaching and performing (range 10–32 years). They were experienced in delivering dance to people living with PD and were accomplished in dance genres such as jazz, ballet, tap, hip hop, Bollywood, salsa, Brazilian dance, Latin, ballroom, and Argentine tango. Their demographic details are summarised in Table 2.

The two physiotherapy (PT) assessors were both experts in movement disorders and registered with AHPRA. Both had bachelor’s and master’s degrees, were aged 30 and 62 years, respectively, and had 40 and seven years of clinical and research experience, respectively. Their demographic details are summarised in Table 2.

### 2.5. Data Management

Electronic copies of consent forms, audio-recordings, and de-identified transcriptions were stored on a secure university research drive with access restricted to the main lead researcher and project manager. Demographic and assessment data were stored in a secure Recap folder accessible to the lead researcher, project manager, and REDCap survey developer.

### 2.6. Intervention

ParkinDANCE Online procedures were described previously [30]. The online classes were delivered one hour, twice per week for 4 consecutive weeks. Because the safety, feasibility, and processes associated with online dancing had not previously been tested, the ethics committee approved a 4-week, 8-session trial as an initial step.

### 2.7. Process Mapping

(i)Mapping contextual factors

Contextual factors that shaped how ParkinDANCE Online were delivered related to (i) the COVID-19 pandemic, which necessitated all participants to work from or stay at home and to access the classes via the internet (ii) our recent systematic review findings on dance for PD [26], which informed the design of the classes, including the content, dosage, intensity, frequency, dance genres, and music genres.

In Melbourne, Australia, the first COVID-19 hard lockdown commenced on 21 March 2020, and the most recent lockdown finished on 22 October 2021. During this time, people were isolated to their home environment, sometimes alone and sometimes only with direct family members. Social isolation was widely reported [53,54], and many people with chronic diseases sought online activities to keep moving [55] or to keep socially connected [56]. When the hard lockdown was lifted, people with chronic conditions were advised to avoid mixing in large community groups and to work at home as much as possible. Dance teachers, physiotherapists, PD industry stakeholders, and the research team were also required to work from home, necessitating fully online project delivery.

(ii)Methods for “Process Analysis of What was Delivered”


**(a) Planning for delivery**


Prior to delivery of this new mode of dance therapy, a steering committee of stakeholders from La Trobe University and Fight Parkinson’s was formed (MEM, SCS, VM). Key considerations included safety, staff training, dance program content, delivery of classes, and how to ensure that people with PD were engaged in co-production. The steering committee obtained ethics approval and appointed project staff. The interventions and procedures were co-designed by the team in conjunction with each individual dancer.

Prior to the dance classes, the project manager, who was a registered physiotherapist, conducted safety screening at home via Zoom^®^. Two experienced movement disorders physiotherapists (PTs) then conducted full assessments online via Zoom^®^. Under the supervision of the chief investigators, the project manager conducted PT assessor orientation and dance instructor education, research project training, and choreography feedback. This ensured that safety protocols, class content, dance routines and assessments were delivered as planned. The chief investigator also checked dance class plans, choreography, and music to ensure alignment with the ParkinDANCE model and to ensure compliance with ethics and copyright regulations.

The team co-designed a range of manuals for dancers (Appendix A), dance instructors (Appendix A), blinded assessors (Appendix A), and process auditors in addition to a videoconference guide (Appendix A). After each class, each dancer and dance instructor completed an online questionnaire via REDCap. Two weeks after the completion of the dance program, each dancer participant was re-assessed by the same movement disorders physiotherapist.


**(b) Management of protocol variations**


We recorded any unintended variations from the planned delivery on the fidelity checklist (Table 3). Variations included a shift of safety at home screening from Fight Parkinson’s to the project manager, and a small number of interactive electronic participant screening forms were replaced by non-interactive forms, which were then filed on the La Trobe University research drive.

(iii)Methods for Recording “What Was Delivered”


**(a) Participant orientation**


The preliminary screening data for each participant was checked by the project manager for eligibility. This included participant training, provision of manuals and resources, and completion of dress rehearsals. These data were recorded in the fidelity checklist (Table 3).


**(b) Pre- and post-intervention blinded assessment**


Each PT assessor was assigned the same six dancer participants for pre- and post-intervention assessments using outcome measures for balance, gait, quality of life, and disability. The PT assessor guidelines were used in conjunction with orientation and a dress rehearsal with the project manager. Each assessor had an orientation meeting with their assigned participants and then scheduled a full online assessment one week before and two weeks after the dance intervention. Results were uploaded into REDCap for further analysis.


**(c) Dance class choreography and music plans**


A dance instructor manual was developed by the research team prior to implementation of ParkinDANCE Online. The manual was used for pre-intervention dress rehearsals between the lead researcher, dance teachers, and project manager. The manual contained class plans (including suggested music and choreography) to be used by the dance instructors to inform the choreography for their classes. Each dance teacher documented the details of each of the classes that they delivered, as well as their impressions of participant performance and engagement. The diaries were stored in REDCap.


**(d) Post-**
**session surveys (dancers and dance instructors)**


Attendance was recorded by the dance instructors. Where a class was missed, due to technology difficulties, it was rescheduled and completed. A post-class online survey was sent to each participant and dance instructor within 24 h of class completion using REDCap questions, which included music and dance enjoyment, adequacy of rest breaks, class structure, and class enjoyment. The dance instructors also completed post-class surveys (eight surveys for each of three dancers) about delivered content, technology problems, safety, and health.


**(e) Post-**
**intervention interviews and focus groups**


Twelve dancers with PD were individually interviewed by the project manager. The zoom interviews were audio-recorded, transcribed verbatim, thematically analysed within an interpretive phenomenological framework, and comprehensively reported [30]. Four dance instructors attended a focus group, and two PT assessors were individually interviewed and analysed with the same methods as the dancers.

(iv)Analysis: ParkinDANCE Online was implemented as planned

The analysis of whether ParkinDANCE Online was implemented as planned, and the extent to which the expected outputs occurred was determined from process mapping as recommended by Antonacci et al. [57]. Steps included data and information gathering from an auditing process, process map generation, and analysis of the extent to which expected processes and outputs occurred.


**(a) Process identification**


The first process pertained to the recruitment protocol, and the second related to the co-designed screening documents. We audited the process by which Fight Parkinson’s performed recruitment and screening using pre-prepared documents that included a letter of invitation, participant information and consent forms, eligibility criteria, doctor screening tool, personal information, medical information, and self-assessment of technology and safety at home. The next process was the development and use of project manuals. Co-designed project manuals included a participant manual, video conferencing guidelines, physiotherapy (PT) assessor guidelines, PT assessor manual, and a dance instructor manual. The participant manual and video conferencing guidelines were distributed to all participants before the intervention commenced. The PT assessor guidelines and manual were used in conjunction with orientation and a dress rehearsal with the project manager. The dance instructor manual was used in conjunction with pre-intervention dress rehearsals with the lead researcher and project manager.


**(b) Fidelity**


Fidelity checking [58,59] was administered by the project manager using co-designed checklists for pre-intervention and post-intervention intervention screening and assessments, weekly attendance and feedback, and variations and adverse events (Table 3). During week three, a member of the research team (JW, SS, or VM) attended assigned classes and completed the fidelity checklist for the intended/actual dancing class (Table 3).

(v)Stakeholder Views

Implementation success was also determined from qualitative analyses of in-depth interviews from key stakeholders. The experiences and perspectives of the dancer participants, PT assessors and dance instructors were explored to gain insight into co-design processes, facilitators, and barriers in design and delivery, as well as suggestions for future design. Individual interviews were conducted with each of the dancer participants and the two PT assessors, and a focus group was conducted with the four dance instructors. Table 4 shows the interview questions. The qualitative analyses were conducted using a phenomenological theoretical framework and inductive thematic analysis [60,61,62].

## 3. Results

Table 1 summarises the PD dancer demographic data. Table 2 summarises the dance instructor and physiotherapist assessor demographic data. The process evaluation and fidelity analysis are presented in Table 3. Table 4 summarises the dancer post-class survey results (Table 5).

*(i)* 
*The contextual factors*


The contextual factors that shaped how ParkinDANCE Online was delivered were predominantly related to the lockdown conditions in Victoria, Australia, associated with the COVID-19 pandemic. This meant that all therapies were required to be delivered digitally. People could not meet face-to-face and were only permitted to leave the home for up to one hour per day. Although our recent global systematic review on dance for PD [26] showed that face-to face and partnered dance classes are enjoyable and beneficial for social interaction, this was not possible in the COVID-19 context. We could only implement dance steps, genres, music, and dosages that were able to be delivered safely and effectively for individuals with PD living at home. To participate, they needed to have good internet access, the ability to operate the technology, and a willingness to do one-on-one classes online.

*(ii)* 
*What interventions and procedures were delivered?*


The interventions and procedures that were delivered are summarised in Table 3. This shows that every class had a 5-min warm-up phase, a 30- to 40-min active dance phase, and a 5-min cool down. Each of the dance instructors delivered eight classes to three participants at mutually agreeable times. In week three, a member of the research team attended the classes as a fidelity assessor (JW: five participants, SS: five participants, VM: two participants). The researchers completed a checklist that included whether the dance instructor asked about the effects of the previous session, safety items, address and phone number, warm-up, active and cool-down content and timing, rest breaks, hydration, screen view, internet issues, and surveys.

*(iii)* 
*Whether ParkinDANCE Online was implemented as planned and the extent to which the expected outputs actually occurred*


The fidelity checklist (Table 3) verified that each of the classes was delivered as per the protocol and choreography, and music suggestions were implemented as per the dance instructor manual. The completed dance instructor post-session surveys also indicated that the classes were delivered as planned, the dancer environment was safe, there were few technological difficulties (some were internet instability or internet speed), the dancers understood instructions, and there were no health issues or adverse events. The overall attendance was 100%. Only one post-session survey was missed by a dancer participant out of a total of eight surveys for each of 12 dancers. The dancer ratings were generally favourable to very favourable. Overall, enjoyment was high, and the perceived benefits from dancing were in balance, flexibility, confidence, and positive interactions with the dance instructor.

*(iv)* 
*Stakeholder views about implementation, barriers and recommendations*



**(a) Dancer participant in-depth interviews**


Overall, the dancers with PD reported being able to master new movements and skills. They advised that being guided by experienced dance instructors helped them to enjoy the classes and learn new skills. They also reported that the type of music and beat helped them to dance. They advised that support and information from the research team helped them to prepare for the online delivery. The results of this analysis are published in Morris et al., 2021 [30].


**(b) Dance instructors focus group**


Four dance instructors attended the audio-recorded focus group, which lasted for 55 min. All dance instructors had a positive experience and felt well-prepared and supported. The following five themes were identified and supported by quotations linked to the transcript: (1) online therapeutic dancing is accessible and enjoyable; (2) safety procedures are essential for online delivery; (3) screening and provision of participant summaries are essential; (4) with training, dance instructors can adapt class content to movement disorders and individual symptoms; and (5) digital delivery requires training and technology support.


**b1. Online therapeutic dancing is accessible**


All of the dance instructors indicated that online delivery facilitated classes that people could access when otherwise unable, whether due to COVID-19 restrictions, medical limitations, geographical location, or transport availability.


*“it was fantastic to make it accessible for people who are regional … clients I had were able to access the technology and use that and navigate their way around it, and I didn’t feel like I needed to compromise the session being an online format”*

*P3, lines 29–35*



*“I’m really happy that the online option was available, and I think that it is great that they can do it from home, and they don’t need to necessarily leave their house”*

*P1, lines 55–58*



*“they don’t necessarily have the means or a person to be able to bring them to class or face-to-face classes, for example, or they just may be a little bit too far for them to be able to travel”*

*P4, lines 90–93*



**b2. Safety procedures are essential for online delivery**


All of the dance instructors appreciated the education about the importance of safety procedures and valued the safety screening.


*“I really appreciated that you did the screening, and also to discuss with them what’s a safe space to work in. I think that was really valuable to have that prior to the sessions”*

*P2, line 117*



*“emphasised on the importance of the room setup … with regards to safety … they need to be free of space, no rugs”*

*P3, lines 602–604*



*“first and foremost, we’ve got to be able to make sure they’re doing safe things”*

*P4, line 314*



**b3. Screening and provision of participant summaries are essential**


The pre-intervention screening procedures and participant demographic data were considered by dance teachers to be helpful for planning, individualising, and progressing the dancing classes.


*“I was really impressed with the screening. The three people that I had were all at different levels … made the modifications and progressions, depending on the client … screening was really valuable”*

*P2, lines 125, 129*



*“I felt like I had enough information from my participants. All three of them were completely different … that conversation prior to our first session on the phone actually helped me get a bit more information … helped me understand where they’re at and what they’ve done, and possibly what they would be capable of doing in class”*

*P1, lines 142–148*



*“the screening was very valuable, to know a little bit of the background. I found that was very important … really helped me understand their ability and their level”*

*P3, lines 164, 171*



*“the screening was very valuable … so that it was very clear for me as to how to move forward and work with each individual”*

*P4, lines 175, 181*



**b4. With training, dance instructors can adapt classes to individual symptoms**


The dance instructors all valued the preparation, education, and support manuals provided by the research team. They were able to tailor classes to the individual and respond to symptom fluctuations.


*“I felt a lot of the pre-planning, recording our sessions and writing—I felt that really helped for me planning, that when we came live, I think that was a really good … I had a lot of guidance and support in the actual preparation of how to deliver a safe class. That was really good. And I felt there was a lot of planning involved and support from the research team”*

*P3, line 434–441*



*“the actual communication that you gave us was really clear, the follow-up was clear”*

*P2, line 412*



*“I really felt supported all the way. The communications were great and just guiding us through the preparations was really great for us to have a clearer idea of what the study would be focusing around and for our own preparation”*

*P4, line 450–452*



*“going back and referring to that manual was quite important”*

*P3, line 509*



*“I would put another kind of element in the class so that I could see how they would cope with that amount of movement … adapt that session to that particular person”*

*P1, line 590*



**b5. Digital delivery requires training and technology support**


Dance instructors reported that online delivery requires a stable internet connection and a device that can be moved and show the entire body. Video conference manuals provided a reference point, and many dancers had access to technology support.


*“the clients I had were able to access the technology and use that and navigate their way around it, and I didn’t feel like I needed to compromise the session being an online format”*

*P2, lines 33–34*



*“they adapted to Zoom. Some knew a lot more about technology than others, but everyone sort of adapted”*

*P3, line 47*



*“all of my students were great on Zoom, and even if they maybe had some troubles every now and then, we kind of had a bit of a discussion about it and almost kind of made it a social element to the session as well”*

*P1, lines 59–61*



*“making sure that you’re tracking and using the (dance instructor and videoconference) manual method to do that. So, I think in planning and to know what you’ve done and to clearly document things in those ways, I found it quite useful”*

*P2, lines 503–505*



**(c) Physiotherapy assessor in-depth interviews**


The two movement disorders physiotherapists who conducted the pre- and post-intervention assessments were individually interviewed in detail. From the de-identified transcribed audio-recordings, two analysts reached a consensus on the following four themes and identified the supporting quotations linked to the transcripts: (1) online dancing can be made safe with participant screening and environment checks; (2) assessor expertise, preparation, guidance, and support are essential; (3) online delivery enables increased access to therapeutic dancing; and (4) online assessment is challenging and requires measurement and technology adaptation.


**c.1 Online dancing can be made safe with screening and environment checks**


The assessors, who were movement disorders physiotherapists, reported that the screening of medical symptoms and home environment could make the online dancing safe.


*“I very much felt it was delivered safely, and I think the particular cohort of participants we had were very appropriate for online delivery”*

*P1, line 66–67*



*“I certainly felt that I got a good idea of their falls risk, level of balance from that (online assessment). I think if there’d been anyone that I’d been really concerned about, it would’ve picked up in the assessment, and I would’ve passed that back to you”*

*P1, lines 128–130*



*“their screening was really, really useful to know that they’d been in a kind of initial safety screen and risk assessment”*

*P1, line 178*



*“they were (well screened). They were all very keen, they were all able to follow instructions, and to engage in the tasks, and were quite appropriate … I don’t recall having any safety concerns”*

*P2, line 365*



**c.2 Assessor expertise, preparation, guidance, and support are essential**


The physiotherapists acknowledged that their expertise in movement disorders was an advantage, and they agreed that the preparation and orientation provided by the research team was valuable.


*“It was like a (pre-assessment) familiarisation … meant you actually met them and got a little bit of a feel, saw them moving a little bit before you were doing the assessment then a few days later … it gave me a really good insight. So that was definitely very valuable to do”*

*P1, lines 83–86*



*“you certainly need physiotherapists that are familiar with working with people with Parkinson’s … also taking into account that kind of holistic approach with cognition and picking up if someone is impulsive”*

*P1, lines 142, 146*



*“that (the assessor manual and videoconference guide) was brilliant. Really useful to have”*

*P1, line 196*



*“I was using every ounce of skill I had to try and help to make the person feel supported and move it through (the assessment)”*

*P2, line 215*



*“I felt I had easy good communication with yourself, or with (the research team)”*

*P2, line 503*



**c.3 Online delivery enables increased access to therapeutic dancing**


The physiotherapists had previously participated in the in-person delivery of therapeutic dancing for PD. They reported that the online option was a necessary alternate for people who could not reach community classes.


*“the idea to trial a feasibility of the online was good. And I think it needed to be done. And I think it’s opened up the doors to delivering programs online, whether that’s dance or other kinds”*

*P1, line 58–60*



*“one of the benefits is it allows people to participate rurally. But I don’t know if many of the participants were drivers. And so that if they’d had to be somewhere, they would have had to have got there either by someone driving them, or by public transport, which again is probably not very accessible”*

*P2, lines 68–72*



*“being in people’s homes was accessibility … they weren’t tired when they got there, they were fresh. For a lot of people, having made the whole effort of getting ready and going out, and getting to somewhere, and parking, and getting in, they’re already a bit tired, for these people who do get tired more quickly”*

*P2, lines 639–640*



**c4. Online assessment requires measurement and technology adaptations**


The physiotherapists were experienced in the assessment of people with PD. They reported barriers to online assessment that included not being able to manually assess strength, balance, and rigidity and adjusting to the visual challenges of digital devices.

### 3.1. Measurement Adaptations


*“I’m used to doing it in person, I know which tests are riskier … you have to tailor that approach online. But certainly, being familiar with which parts of the assessment are the riskier part … you just have to think about it a little bit more (online)”*

*P1, lines 153, 157, 166*



*“I adapted the assessment to try and minimise how much the participant was moving the laptop. I did all the assessments where they were in where I needed them in full standing”*

*P1, lines 205–206*



*“you can’t recreate it (push/pull test) over Zoom, because you can’t actually pull—you wouldn’t—you can’t get someone to unbalance themselves … with the rigidity testing, again, you need to feel the movement”*

*P1, line 223, 225*



*“everyone worked out a way to have a runway that was visible, so that they had to walk away from the screen and then back towards it again. I found it much harder to pick up gait abnormalities than I would have normally”*

*P2, lines 110–112*



*“keep in mind when they’re administrating it (UPDRS assessment) because, I think the motor part takes longer online than it does in real life … you haven’t seen the person walk in, you haven’t seen their arms do anything. You have to make the person do it all”*

*P2, lines 251, 254–256*


### 3.2. Technology Adaptations


*“it (pre-assessment meeting) helped just to sort of calm down any anxiety over the technical side of using Zoom and just get them, I guess, used to the format”*

*P1, lines 101, 105–107*



*“I don’t think you could do it with a desktop because you can’t get that movement … I think if we tried using mobile phones, there would be a little bit of an issue of safety with them trying to hold the phone and position the phone, because it’s just not as stable as a laptop”*

*P1, lines 315, 318*



*“if the lighting was good and the webcam was good, there were no issues. But if their webcam wasn’t great, it didn’t feel as accurate as assessing in person”*

*P1, line 555*



*“another challenge I think was the mental concentration. I think we all know that doing a Zoom interview is quite challenging cognitively”*

*P2, line 205*


## 4. Discussion

Online dance therapy offers great potential for movement patterns and associated choreographies to be adapted to individual needs. Whereas physiotherapy has traditionally optimised motor control, movement in the context of dancing affords opportunities for creative expression, and enjoyment, in addition to increasing the amount and range of structured physical activities. This process evaluation showed that the ParkinDANCE Online classes quickly reached the target group of people living with idiopathic PD who were living at home and seeking to become involved in structured physical activity. Despite the COVID-19 lockdown that was in place throughout the entire period of this trial [34], the online mode of delivery was feasible, safe, and valued by the participants. The project activities reached all target groups (patients, dance teachers, and trial assessors), and there were no adverse events. In large part, ParkinDANCE Online was implemented as intended and the trial had strong fidelity. The only aspect that required adjustment was the participant screening process. It quickly became apparent that screening needs to be simple and easy to document, so that community agencies and clinicians could rapidly admit a person with PD to ParkinDANCE Online.

The key stakeholders were satisfied with the project design and delivery. Of note, the people living with Parkinson’s reported satisfaction with the online medium and enjoyment of the ParkinDANCE program. They found the online classes to be engaging, enjoyable, and therapeutic. The dance teachers advised that receiving education about PD was very helpful, as well as how to modify choreography and music to adjust for movement disorders and non-motor symptoms. The physiotherapy assessors reported that it was essential to have access to comprehensive reporting templates and training in how to administer online formats of the UPDRS and other tools. Fight Parkinson’s, the key industry partner, cited the importance of regular, well-organised project meetings and the detailed tracking of project deliverables. They reported that all materials, information, and resources were suitable for the target audience.

Stakeholder views showed that implementation success was largely related to the co-design and co-production of the project. This is in agreement with the recent research on the value of co-design in rehabilitation [63,64,65,66]. A recurring theme was that the implementation of a new therapeutic intervention and delivery mode needs genuine input from people with PD, researchers, health professionals, and industry partners. The main barriers encountered related to the lack of access for people with more severe levels of disability, as well as the short duration of the program, which was 4 weeks for this pilot trial. There was consensus that the program should continue in its entirety, although delivered for 12 weeks or more, rather than only 4 weeks.

The other recommendations evolving from the trial were: (i) participant manuals are helpful for supporting implementation and should be professionally produced and made available online; (ii) reliable, safe, and valid methods for administering the UPDRS and other measurement tools online need to be confirmed; (iii) health care professionals and dance instructors need education and training in how best to deliver online dance classes for people with progressive neurological conditions such as PD; (iv) people living with PD need access to reliable technologies, such as a stable internet connection and a digital device such as laptop, as well as digital literacy and capability.

Although this process evaluation did not aim to investigate the factors associated with the demand and supply of participant recruitment, this can be influenced by factors such as the methods of referral and the availability of professionals to recruit, screen, enrol, and teach the participants. In other contexts, such as singing [30], it has been observed that people with Parkinson’s disease are not always convinced to participate in arts-based interventions, until it is “prescribed” by their doctors or other health professionals. When the ice was broken, arts-based interventions could be very successful, with long-term participation [30]. A challenge that can then arise is that referral and demand for participation can outstrip the supply of dance teachers and physiotherapists trained in the safe delivery of online dancing for people with PD. For complementary therapies for PD with rapidly growing enthusiasm, such as boxing [17], singing [30], and dancing [20], there will be a need to identify local strategies to deal with the demand supply relationship and the potential necessity of rapidly training a skilled workforce to meet growing demand.

There were several limitations of this trial. Although it is one of the first controlled trials of dancing online for PD, it was a phase I trial. A need exists for a randomised controlled trial to reduce sources of bias. We were not able to control PD medications during the trial, and variations in medication could have increased the amount of variability in results. An important limitation is that the age range inclusion criteria was limited to adults aged 18–65, and older adults were excluded. Further research is needed to adapt the program for older adults with PD. In addition, we excluded people with comorbidities; future trials need to test whether ParkinDANCE Online is safe and effective for people with PD who also have other chronic conditions, given that most older people have two or more chronic conditions. Future digital innovations and virtual reality are likely to afford an array of advanced technologies to give participants greater access and opportunities to engage in art health therapies from geographically wide-spread communities.

## 5. Conclusions

The success of online dancing for PD was related to a strong theoretical framework supporting dance therapy design, close adherence to pre-determined interventions and procedures, and incorporating stakeholders in the design and implementation of this innovative and accessible method of online dancing.

## Figures and Tables

**Figure 1 healthcare-11-00604-f001:**
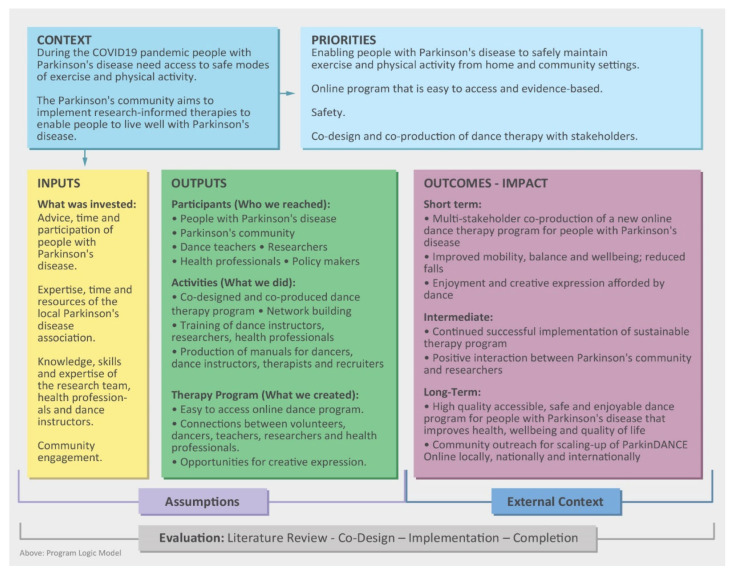
Key Inputs, Outputs, and Context Supporting Program Logic Model.

**Table 1 healthcare-11-00604-t001:** Dancer participant demographic data.

	Interview Time (mins)	Age	Sex	Year PD Diagnosis	H-Y Score	Work Status	Medication	Co-Morbidities	Fall Last 6 Months	Dance Experience
1	29.5	63	M	2018	1	F/T	Madopar, Avanza	Anxiety	No	Last 6 months online (zoom) mixed genre dance
2	48	34	F	2017	3	P/T	Lyrica, Amantadine Hydrochloride	N/A	2	Physical culture and ballet (child)
3	32	64	F	2018	2	Retired	Neupro, Rotigotine	N/A	No	Ballroom, disco
4	28.4	59	F	2020	1	P/T	Madopar	N/A	3	7 years Zumba instructor
5	36.5	59	F	2019	2	Retired	Levodopa	Arthritis (hip), LBP	No	PD Warrior at home, walking
6	36	56	F	2010	2	P/T	Levodopa, Azilect, Sifrol, Thyroxine	Asthma, Hemi-thyroidectomy	No	Ballet—age 10–17, tango, disco, ceroc, zumba classes 2/week, PD Warrior
7	27	61	F	2017	2	Retired	None	Arthritis (neck), osteopenia, R shoulder replacement	No	None
8	23	60	F	2011	1	Retired	None	N/A	No	Ballroom 20 years ago
9	40.5	63	M	2013	2.5	Retired	Levodopa, Azilect, Lipitor, Atacand, Sifrol	Hypertension, Hypercholesteremia	No	None
10	34.5	65	F	2014	3	Retired	Levodopa, Azilect, Neupro patch, Mirtazapine	Depression, Anxiety	1	4 years tango for PD
11	27	59	M	2019	2	F/T	Levodopa	LBP	No	None Zoom Pilates, brisk walking
12	33	57	F	2020	1	Retired	Levodopa, APO-Ezetimibe, Levothyroxine Sodium 125 mg, Gabapentin 300 mg	Dizziness, Hypothyroidism	No	General dancing to live music, walking 5 km/day

**Table 2 healthcare-11-00604-t002:** Dance instructor (DI) & physiotherapist assessor (PT) demographic data.

**Participant** **DI**	**FG Time (Minutes)**	**Age**	**Gender**	**PD Training**	**Years of Experience**	**Qualifications**	**Dance Genre**
1	55.5	36	F	Y	32; 20 years performing	Cert 3 and 4 Musical Theatre/Performing Arts: 2002 RAD Ballet, CSTD Jazz and Ballet: 1995, 2005	Jazz, Ballet, Tap, Hip Hop, Bollywood, Salsa, Brazilian Dance, Latin and Ballroom, Argentine Tango
2	55.5	38	M	Y	17.5	Performing Arts Course Year: 2007–2009 Cert 1 and 2 in Fitness: 2001; Certificate 3 and 4 in Fitness: 2018 Gyrotonic Method Certified: 2018; Gyrokinesis Licensed Instructor: 2020; Jumping Stretch Board Specialised equipment trainer: 2019; Cert 3 and 4 in Training and Assessment: 2019; BSci–2 years	Jazz, Ballet, Tap, Hip Hop, Bollywood, Salsa, Brazilian Dance, Latin and Ballroom, Argentine Tango
3	55.5	45	F	Y	15	Finalist- Asian Championships 2016, 2019 Tango for Parkinson’s (Tango Esencia Studio)	Tango
4	55.5	56	F	Y	10 Choreographer, Performer, Instructor	Movement Educator—Modalities—Nia Dance Technique Black Belt Instructor, Ageless Grace Body and Brain Movement program Trainer/Educator, Tai chi/Qigong Instructor, Fitness Instructor	Tango, Latin, Salsa, Ballroom, Jazz
**Participant** **PT**	**FG time (minutes)**	**Age**	**Gender**	**PD Training**	**Years of experience**	**Qualifications**	**Expertise**
1	46	30	F	Y	7 Public and Research	BSc Physio (Hons) Masters Public Health	Movement Disorders Physiotherapist
2	61	62	F	Y	40 Public and private	BAppSci (Physio) MPhysio (Research)	Movement Disorders Physiotherapist

**Table 3 healthcare-11-00604-t003:** Fidelity checking and process analysis.

Theoretical Elements (Area to Measure)	Research Questions/Aims	Data Sources and Data Collection Methods																
		FC	PDV	S/R	DS	RC	M	SC	RA	PTA	DI	PD	DQ	DIQ	DCA	DII	DIFG	PTAI
**Ethics & permissions**																		
Ethics approval	Was human ethics approval obtained?	✓	✓		✓		✓	✓										
Ethics variations	Were approvals for variations obtained?	✓	✓		✓		✓	✓										
PROMS licences	Licences obtained outcome measures?	✓			✓		✓	✓										
Trial registration	Was the trial registered?	✓						✓										
**Recruitment**																		
Eligibility criteria	Were *a priori* eligibility criteria stipulated?	✓	✓	✓	✓	✓		✓										
Representative sample	Were *a priori* recruitment procedures used to attract individuals to the intervention?	✓	✓	✓	✓			✓										
Document development	Were project documents co-produced?	✓	✓	✓	✓			✓	✓									
Protocol: advertising, recruitment & screening	Did the recruiters follow protocol?	✓	✓	✓	✓			✓										
Documents completed	Were documents checked for accuracy and completeness?	✓	✓	✓	✓			✓										
**Data management**																		
Research drive folders	Were secure folders created on research drive for data storage?	✓			✓			✓										
Research drive access	Was access to research drive folders restricted to designated team?	✓	✓		✓			✓	✓									
REDCap data templates	Were secure REDCap data capture templates created?	✓			✓	✓		✓	✓									
REDCap surveys	Were participant surveys created & stored in REDCap?	✓			✓	✓		✓	✓									
**Manuals**																		
Participant manual	Was dancer participant manual co-produced?	✓	✓		✓		✓	✓	✓	✓	✓							
Videoconference guide	Was videoconference guide co-produced?	✓	✓		✓		✓	✓	✓	✓	✓							
PT Assessor guide	Physiotherapy Assessor guide co-produced?	✓	✓		✓		✓	✓	✓	✓								
PT Assessor manual	Physiotherapy Assessor manual co-produced?	✓	✓		✓		✓	✓	✓	✓								
Dance instructor manual	Dance Instructor manual co-produced?	✓	✓		✓		✓	✓			✓							
Fidelity templates	Were fidelity checklists co-produced for fidelity assessments?	✓	✓		✓		✓	✓										
**Training**																		
DI education (zoom)	Was weekly education provided via zoom with the project manager?	✓	✓		✓						✓						✓	
DI choreography	Did the Dance Instructors and project manager co-design choreography?	✓	✓		✓		✓				✓						✓	
DI dress rehearsal	Did each Dance Instructor have a dress rehearsal (zoom) with the project manager or lead researcher?	✓			✓		✓	✓			✓						✓	
PTA orientation	Did each PT assessor have orientation (zoom) with the project manager?	✓			✓		✓	✓		✓								✓
PTA dress rehearsal	Did each PT assessor have an assessment dress rehearsal (zoom) with the project manager?	✓			✓		✓	✓		✓								✓
**Assessment**																		
Pre-intervention	Was pre-intervention assessment conducted for each participant by a PT assessor?	✓			✓	✓				✓		✓				✓		✓
Post intervention	Was post-intervention assessment conducted for each participant by a PT assessor within 2 weeks following the last class?	✓			✓	✓				✓		✓				✓		✓
**Surveys**																		
Dancers, post class	Did each dancer complete an online REDCap survey after each class?	✓			✓	✓			✓			✓	✓				✓	
Dance Instructor – post class	Did each Dance Instructor complete an online survey after each class?	✓			✓	✓			✓		✓			✓			✓	
Dance Instructor diary	Did each Dance Instructor complete a diary for each class?	✓			✓		✓				✓						✓	
Project manager phone calls	Did the project manager complete weekly phone calls to each dancer?	✓			✓													
**Intervention adherence**																		
Attendance	Did each dancer attend all classes?	✓			✓	✓					✓	✓	✓	✓	✓	✓	✓	
Choreography as planned	Did the Dance Instructors deliver the planned choreography?	✓			✓	✓	✓				✓			✓	✓		✓	
Week 3 fidelity check	Did the fidelity checker observe that the class was delivered as planned?	✓	✓		✓		✓	✓							✓			
Class co-design	Were the classes co-designed for music & dance preferences?	✓			✓						✓	✓	✓	✓	✓	✓	✓	
Dosage & delivery	Were the intervention components implemented as often and for as long as planned?	✓			✓	✓					✓			✓	✓		✓	
Content	Was each intervention components implemented as planned?	✓			✓		✓				✓			✓	✓		✓	
**Qualitative results**																		
Participant responsiveness	Were the participants engaged with the interventions?	✓				✓		✓		✓	✓	✓	✓	✓	✓	✓	✓	✓
Participant experience	Were the participants satisfied with the interventions?	✓				✓		✓		✓	✓	✓	✓	✓	✓	✓	✓	✓
Participant experience	Were there barriers to implement the interventions?	✓				✓		✓		✓	✓	✓	✓	✓	✓	✓	✓	✓
Participant experience	Were there facilitators to implement the interventions?	✓				✓		✓		✓	✓	✓	✓	✓	✓	✓	✓	✓
Participant experience	Were support strategies implemented?	✓				✓		✓		✓	✓	✓	✓	✓	✓	✓	✓	✓
**Adverse events**																		
Safety checks	Was safety screening at home conducted before the intervention?	✓		✓	✓								✓	✓	✓	✓	✓	✓
Safety checks	Was safety at home checked at each class by the Dance Instructor?				✓								✓	✓	✓	✓	✓	
Adverse events	Were there any adverse events?	✓			✓								✓	✓	✓	✓	✓	

Legend: ✓successfully completed 100%, Grey cell: N/A, FC: Project manager fidelity checklist, PDV: Parkinson’s Victoria, S/R: Screening & recruitment, DS: Document storage (research drive), RC: REDCap (data & surveys), M: Manuals, SC: Steering committee, RA: Research analyst, PTA: Physiotherapy assessor, DI: Dance instructor, PD: PD participant, DQ: Dancer questionnaire, DIQ: Dance instructor questionnaire, DCA: Dance class audit, DII: Dancer interviews, DIFG: Dance Instructor Focus Group, PTAI: PT assessor interviews.

**Table 4 healthcare-11-00604-t004:** Interview and focus group questions.

**Participant interviews**	
1. What did you think about the dance exercises? Did you enjoy or not enjoy the dance steps? Did you enjoy or not enjoy the music? Were there any changes to your balance, walking or quality of life?	2. What is your opinion about the online presentation format? Was it easy or hard to connect and participate? What did you think about the instructors? Did you prefer fixed time or pre-recorded? 3. Were there any environmental problems? Did you feel safe? 4. What did you like or not like about the classes? Do you have any suggestions for changes to the class content or presentation?
**Dance instructor focus group**	
1. What is your opinion about the online presentation format? Were there any barriers to implementation? What did you think about the participants’ capability? Did you prefer a negotiated time? Did you have adequate information about the participants? What do you think about a pre-recorded intervention?	2. Was the environmental and safety screening adequate? Were there any safety issues? Were there any technical problems? Were you confident about safety and lines of communication? 3. Do you have any ideas about individual or group delivery? 4. What do you think about the dance instructor and video-conferencing manuals and pre-intervention preparation? Do you have any suggestions for class content or and delivery?
**Physiotherapy assessor interviews**	
1. What is your opinion about the online assessment format? Were there any barriers to implementation? What did you think about the participants’ capability? Did you have adequate information about the participants? Do you have any recommendations for future design or implementation?	2. Was the environmental and safety screening adequate? Were there any safety issues? Were there any technical problems? Were you confident about safety and lines of communication? 3. Do you have any ideas about individual or group delivery of dance classes? 4. What do you think about the PT assessor guidelines and assessment manuals and video-conferencing manual? 5. What do you think about your pre-intervention preparation and support through the project?

**Table 5 healthcare-11-00604-t005:** Dancer post class surveys (score per question 1 (strongly disagree)—5 (strongly agree), mean total for each question).

Participant	1. Enjoyed Dance Steps	2. Enjoyed Music	3. Rest Breaks Appropriate	4. Mix of Sitting and Standing	5. Overall Enjoyment	Comments
1	5	5	4.5	5	5	Fun, assist flexibility, coordination. Good blend of music. Enough rest breaks. Enjoyed increased challenge over the 4 weeks. Good feedback and support.
2	5	4.5	3	5	5	Increased confidence with achieving movements. Fun and encouraging instructor.
3	4.75	4.75	4.75	4.75	4.75	Needed more high energy dancing. Great teacher.
4	5	4	5	5	4.5	Increased confidence and coordination. Engaged with instructor—patient and respectful.
5	4.5	4.5	4.5	4.5	4.5	Fun but need more challenge in standing rather than sitting. Enjoyable and liked variety. Challenge increased over time.
6	5	4	4.5	5	5	Repetitive but helped to remember steps. Good tuition. Zoom a good option but would prefer in-person.
7	5	4	5	5	5	No comments.
8	4	4	4	4	4	Enjoyable and not too strenuous. Great instructor and very positive.
9	5	5	5	5	5	Great that dance can be an adaptable activity. Increased confidence and less reliance on the chair for balance.
10	4.5	5	5	5	5	Great variety of dance steps—not too late in life to learn. Dance instructor was intuitive.
12	5	4.75	4.5	5	5	Enjoyed instructor’s energy. Fun and a good challenge. Did not need so many breaks.
13	4.5	4.75	4.5	4.75	4.5	Enjoyed very much—lots of fun.

## Data Availability

Data supporting reported results can be found within this manuscript.

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
