# Peer review of "Dancing for Parkinson’s Disease Online: Clinical Trial Process Evaluation"

_healthcare, 2023, doi:10.3390/healthcare11040604_

Round 1

Reviewer 1 Report

This is an interesting manuscript on delivering a research based dance program for Parkinson's disease online. The manuscript was well written and adds to the literature gap on complimentary therapies for persons with Parkinson's disease. I have no comments and recommend accept.

The manuscript addresses a gap in the literature and provides information on transitioning dance for PD to an online platform. While the purpose of the manuscript is not strictly research, it does provide valuable information on methods, data, and response of participants to demonstrate the feasibility of the project. The authors state their purpose, and provide the data to support the purpose of the manuscript.

The background/introduction is well formulated and written. Methods are easy to follow and results are reported well. There is a smaller discussion section, but that would be expected for a manuscript in this format. Finally, the authors do share the data. To my knowledge, there is little data shared in this area of research. This would set a precedence for research in this area that would move the field forward.

Author Response

Responses to Reviewer 1

Reviewers comment 1: This is an interesting manuscript on delivering a research-based dance program for Parkinson's disease online. The manuscript was well written and adds to the literature gap on complimentary therapies for persons with Parkinson's disease. I have no comments and recommend accept. The manuscript addresses a gap in the literature and provides information on transitioning dance for PD to an online platform. While the purpose of the manuscript is not strictly research, it does provide valuable information on methods, data, and response of participants to demonstrate the feasibility of the project. The authors state their purpose, and provide the data to support the purpose of the manuscript. The background/introduction is well formulated and written. Methods are easy to follow and results are reported well. There is a smaller discussion section, but that would be expected for a manuscript in this format. Finally, the authors do share the data. To my knowledge, there is little data shared in this area of research. This would set a precedence for research in this area that would move the field forward.

Response: Thank you for these very positive comments.

Reviewer 2 Report

This study conducted a process evaluation for a clinical trial on online dancing for people living with Parkinson's disease (PD). The program, called "ParkinDANCE Online," was co-produced by people with PD, healthcare professionals, researchers, dance instructors, and a PD organization. The evaluation included a stakeholder steering group to oversee program design, co-design of online classes, education of dance teachers, fidelity checking, online surveys, and post-trial focus groups and interviews with participants. The results showed that the program was feasible to deliver online and was well-received by participants and dance teachers, with no attrition or adverse events and strong program fidelity. The study also found that safety was facilitated by careful screening and a home safety checklist. In terms of the topic of the research, it overlaps with the team's previous publications, please explain the main differences between the two and the importance of this publication: "Online Dance Therapy for People With Parkinson's Disease: Feasibility and Impact on Consumer Engagement" Neurorehabil Neural Repair . 2021 Dec;35(12):1076-1087. doi: 10.1177/15459683211046254.

Author Response

Responses to Reviewer 2

Reviewer 2, comment 1: This study conducted a process evaluation for a clinical trial on online dancing for people living with Parkinson's disease (PD). The program, called "ParkinDANCE Online," was co-produced by people with PD, healthcare professionals, researchers, dance instructors, and a PD organization. The evaluation included a stakeholder steering group to oversee program design, co-design of online classes, education of dance teachers, fidelity checking, online surveys, and post-trial focus groups and interviews with participants. The results showed that the program was feasible to deliver online and was well-received by participants and dance teachers, with no attrition or adverse events and strong program fidelity. The study also found that safety was facilitated by careful screening and a home safety checklist.

Response: Thank you for these very positive comments.

Reviewer 2, comment 2: In terms of the topic of the research, it overlaps with the team's previous publications, please explain the main differences between the two and the importance of this publication: "Online Dance Therapy for People With Parkinson's Disease: Feasibility and Impact on Consumer Engagement" Neurorehabil Neural Repair. 2021 Dec;35(12):1076-1087. doi: 10.1177/15459683211046254.

Response 2: Thank you – this unique manuscript does not contain any overlapping data with Neurorehabil Neural Repair. 2021 Dec;35(12):1076-1087.  Whereas that publication reported data on the feasibility of conducting online dancing classes for people living with Parkinson’s disease, this manuscript publishes a process evaluation of that trial.

In the context of a clinical trial, a process evaluation analyses whether predetermined research program activities have been implemented as planned and whether or not they resulted in particular outputs. According to French et al. (2022) process evaluations are very helpful to include in clinical trials because they enable greater understanding of the mechanisms of complex interventions by investigating implementation processes, the context of trial delivery and the mechanisms of impact. They yield data that can help interpret outcomes as well as assisting the implementation of findings into future clinical practice (French et al. 2022). 

We have added a new short paragraph (stating the above) in the methods section of the article to explain this to the readers.

Reviewer 3 Report

The paper that the authors have put together quite nicely captures the potentials, intricacies, complexities, and implications of online therapeutic movement interventions for people with PD. I applaud to the thoughtful design of a process evaluation protocol based on responses from the various stakeholders.

Of course, there remain gaps, uncertainties, and aspects which remain a bit vague.

1. Recruitment. The authors mention convenience sampling. Now, is it possible to measure the success or failure of recruitment efforts? In other contexts, it has been observed that PD-patients were not convinced to participate in arts-based interventions until it was "prescribed" in mild ways by their doctors. However, reports suggest that when the ice was broken, arts-based interventions such as singing, could be very successful, too.

What, if the word is spread that dance/music is helpful and 99% of patients are eager to participate? Would the supply of dance/physiotherapists be great enough to meet the demand? What if recruitment required selection, i.e. more strict inclusion criteria, because capacities are limited? That seems, of course, of no concern for the present study, but, if process evaluation is thought a bit further, it may entail also strategies to deal with the demand/supply relationship and the potential necessity of finding solutions in the case of imbalances, which could be anticipated;

2. Covid-19. Social distancing is a burden, for sure, but a telemedicine increasingly a necessity in geographically wide-spread communities. That brings advanced technologies of virtual reality/metaverse into question. Zoom is certainly a first step, but did the authors consider to what extent  the ongoing developments could change the picture and which areas of present concern could benefit (on both sides instructors and clients)?

3. Personalized medicine. It seems like a huge asset that movement patterns/choreographies can be adapted and must be adapted to individual needs. But I would think that adaptation is what characterized physiotherapy, in general. In what ways does the current implementation hold any specialities in terms of adaptation of the dance program content to individual needs?

4. Content. There is little information about the choreography and music. It would be nice to have examples of those materials. Could the authors give hints about some basic features of choreographies, musical tempi, genres etc. I would not expect Rock'n'roll with "death jumps" over the back of the partner, but having no information at all is clearly not too enlightening about the content that was actually delivered. If video-fragments, excerpts of choreopgraphies from the materials being used could be incorporated, that would enormously help those who would consider joining the band wagon.

I voted an "accept in present form". Yet, I hope that the authors pick up my enthusiasm and address some of my recommendations to ammend their manuscript in some parts in response to the minor issues outlined above.

Author Response

Dear Reviewer,

comment 1: The paper that the authors have put together quite nicely captures the potentials, intricacies, complexities, and implications of online therapeutic movement interventions for people with PD. I applaud to the thoughtful design of a process evaluation protocol based on responses from the various stakeholders.

Response 1: Thank you for these positive comments.

Reviewer 3, comment 2: Recruitment. The authors mention convenience sampling. Now, is it possible to measure the success or failure of recruitment efforts? In other contexts, it has been observed that PD-patients were not convinced to participate in arts-based interventions until it was "prescribed" in mild ways by their doctors. However, reports suggest that when the ice was broken, arts-based interventions such as singing, could be very successful, too. What, if the word is spread that dance/music is helpful and 99% of patients are eager to participate? Would the supply of dance/physiotherapists be great enough to meet the demand? What if recruitment required selection, i.e. more strict inclusion criteria, because capacities are limited? That seems, of course, of no concern for the present study, but, if process evaluation is thought a bit further, it may entail also strategies to deal with the demand/supply relationship and the potential necessity of finding solutions in the case of imbalances, which could be anticipated?

Response 2: Thank you for these insightful comments about recruitment. Although this manuscript did not aim to measure the success or failure of recruitment efforts or how to deal with future issues of demand and supply of services, we agree that it is important to raise this issue in the Discussion. We have therefore added the following new short paragraph in the Discussion, based on these important comments by Reviewer 3:

“Although this process evaluation did not aim to investigate the factors associated with demand and supply of participant recruitment, this can be influenced by factors such as the methods of referral and the availability of professionals to recruit, screen, enrol and teach the participants. In other contexts such as singing, it has been observed that people with Parkinson’s disease are not always convinced to participate in arts-based interventions until it is "prescribed" by their doctors or other health professionals. When the ice was broken, arts-based interventions could be very successful, with long term participation. A challenge that can then arise is that referral and demand for participation can outstrip the supply of dance teachers and physiotherapists trained in the safe delivery of online dancing for people with PD. For complementary therapies for PD with rapidly growing enthusiasm such as boxing, singing and dancing, there will be a need to identify local strategies to deal with the demand supply relationship and the potential necessity of rapidly training a skilled workforce to meet growing demand”.

2. Covid-19. Social distancing is a burden, for sure, but a telemedicine increasingly a necessity in geographically wide-spread communities. That brings advanced technologies of virtual reality/metaverse into question. Zoom is certainly a first step, but did the authors consider to what extent the ongoing developments could change the picture and which areas of present concern could benefit (on both sides instructors and clients)?

Response 2: Thank you for this suggestion. We have added two new sentences in the revised Discussion to reinforce this point: “Although this trial used Zoom to deliver online therapy, future digital innovations and virtual reality are likely to afford a great array of advanced technologies to give participants greater access and opportunities to engage in arts health therapies from geographically wide-spread communities.”

3. Personalized medicine. It seems like a huge asset that movement patterns/choreographies can be adapted and must be adapted to individual needs. But I would think that adaptation is what characterized physiotherapy, in general. In what ways does the current implementation hold any specialities in terms of adaptation of the dance program content to individual needs? 

 Response 3: Thank you for this suggestion. We have added two new sentences in the revised Discussion to say: “Online dance therapy offers great potential for movement patterns and associated choreographies to be adapted to individual needs. Whereas physiotherapy has traditionally optimised motor control, movement in the context of dancing affords opportunities for creative expression and enjoyment, in addition to increasing the amount and range of structured physical activities”.

4. Content. There is little information about the choreography and music. It would be nice to have examples of those materials. Could the authors give hints about some basic features of choreographies, musical tempi, genres etc. I would not expect Rock'n'roll with "death jumps" over the back of the partner, but having no information at all is clearly not too enlightening about the content that was actually delivered. If video-fragments, excerpts of choreographies from the materials being used could be incorporated, that would enormously help those who would consider joining the band wagon.

Response 4: Thank you for this suggestion. In the methods section we have added some more details about the choreography and music. We were not granted ethics approval to video the participants, so on this occasion that cannot be provided. To address this issue, we have set up a new “Figshare” Public Access File Share Repository and in there we have put the following manuals which contain all of this information:

Manual for Dance Instructors (including choreography and music)

Participant Guide

Video Conference Guide for Dance Exercises for Parkinson’s Rehabilitation (Online)

Assessor Guidelines 

Reviewer 4 Report

Dear authors.

I appreciate the opportunity to review the work you have sent, and I praise your effort in carrying out the therapeutic proposal. But I regret to inform you that there are issues that should be corrected and improved so that this work could be accepted.

From the first moment I thought that it will be a quantitative work, but the development of it has made me verify that it is a confusing methodology, being a qualitative study.

Below I list some suggestions for improving the manuscript.

- Abstract: it must be restructured to make it concise and clarifying content.

- Keywords: excess of words, excess of some that I consider are not related to the study.

Introduction:

- you should not say "our review (26)". Is not correct.

- References of line 59 your citation is incorrect.

- line 68: what is MEM?

- reference 30 not adequate and is poorly written in the bibliography

- figure 1 is extensive and not adequate. You have to be more concise (it is not a poster or conference presentation, so the presentation of data in an article must be different).

- Line 117: what is VM?

At various times in the study, reference is made to a previous study (reference 30), but this should not be the case, because this study must show different results or at least be different from the previous one.

Material and methods:

-It is too extensive and does not meet the specificity criteria. Too detailed and extensive.

-Not very important (I consider) the qualifications shown in table 2.

- Table 3 is redundant.

- The methodology followed in the study should be considered from the beginning (serious flaw).

Very little discussion section.

Author Response

Dear Reviewer,

I appreciate the opportunity to review the work you have sent, and I praise your effort in carrying out the therapeutic proposal. Below I list some suggestions for improving the manuscript.

- Abstract: it must be restructured to make it concise and clarifying content

Response 1: We have edited the Abstract to make it clearer and more concise

- Keywords: excess of words, excess of some that I consider are not related to the study.

Response 2: We have removed 2 keywords

Response 3: We have made the following changes: Introduction: - you should not say "our review (26)". Is not correct. (We removed the word “our”) - Reference in line 59 your citation is incorrect. (We checked this and the correct reference is as follows, hence we left it: Emmanouilidis, S., et al., Dance Is an Accessible Physical Activity for People with Parkinson's Disease. Parkinsons Dis, 2021. 2021: p. 7516504.

  • line 68: what is MEM? (This is my initials Meg Elayne Morris hence shall stay the same)
  • reference 30 not adequate and is poorly written in the bibliography. You are correct in saying this and we have removed that incorrect reference.

 Figure 1 is extensive and not adequate. You have to be more concise (it is not a poster or conference presentation, so the presentation of data in an article must be different).

Response 4: Thankyou for this comment. Figure 1 is an important diagram to show all of the elements of our process evaluation. Given this feedback we have moved Figure 1 into a new Figshare Public File Sharing Repository overseen by La Trobe University and given the readers the link to this resource.

- Line 117: what is VM? (Response: Victor McConvey, one of the authors, which we have left)

At various times in the study, reference is made to a previous study (reference 30), but this should not be the case, because this study must show different results or at least be different from the previous one. Response: thankyou we have removed these cross references if unnecessary.

Material and methods:

-Not very important (I consider) the qualifications shown in table 2. Response: The authors believe that this Table has important Process Evaluation details, so we have respectfully retained table 2.

- Table 3 is redundant. Response: The authors believe that this Table has important Process Evaluation details, so we have respectfully retained table 3.

- The methodology followed in the study should be considered from the beginning.

Response: Thank you for this comment – we have added more details on the methodology for the process evaluation from the beginning, including the following new text:

“In the context of a clinical trial, a process evaluation analyses whether predetermined research program activities have been implemented as planned and whether or not they resulted in particular outputs. According to French et al. (2022) process evaluations are very helpful to include in clinical trials because they enable greater understanding of the mechanisms of complex interventions by investigating implementation processes, the context of trial delivery and the mechanisms of impact. They yield data that can help interpret outcomes as well as assisting the implementation of findings into future clinical practice (French et al. 2022). 

For more details please see the revised version manuscript. 

Round 2

Reviewer 4 Report

The corrections made are appreciated, showing an improvement in the initial text.